# A longitudinal pilot study in pre-menopausal women links cervicovaginal microbiome to CIN3 progression and recovery

Cristiana Banila[1,5], Efthymios Ladoukakis [1,5], Dorota Scibior-Bentkowska[1],
Leandro Rodrigues Santiago[2], Caroline Reuter[3], Michelle Kleeman [4] & Belinda Nedjai [1] ✉

Increasing evidence suggests vaginal dysbiosis is associated with persistent high-risk human papillomavirus (hrHPV) infection and cervical intraepithelial neoplasia (CIN) development. In this pilot longitudinal study, we investigate the potential of vaginal microbiome biomarkers to predict CIN3 development in hrHPV-positive (hrHPV+) women of reproductive age and assess loop electrosurgical excision procedure (LEEP) outcomes.

Fifty-nine non-menopausal women 20–53 years old, with normal cytology, were selected from the ARTISTIC trial and followed up twice over six years. Vaginal microbiome was analysed by 16S rRNA sequencing. HrHPV+ women with CIN3 showed a significant overrepresentation of *Sneathia amnii*, *Megasphaera genomosp.*, *Peptostreptococcus anaerobius* and *Achromobacter spanius* ($p < 0.05$). Successfully LEEP-treated hrHPV-negative women exhibited increased *Lactobacillus* species, especially *Lactobacillus gasseri*. Additionally, *Lactobacillus helveticus*, *suntoryeus* and *vaginalis* showed a potential protective role against CIN3 development.

These unique microbial biomarkers associated with CIN3 development and recovery following LEEP treatment bring new insights into the vaginal microbiome's role on disease progression.

Persistent high-risk human papillomavirus (hrHPV) infection can lead to cervical intraepithelial neoplasia (CIN) and ultimately to cervical cancer[1]. Despite the successful development of an HPV vaccine, women in their reproductive years are at higher risk of acquiring hrHPV, and cervical cancer remains the 4th most common cancer among women globally, affecting mostly low and middle-income countries[2]. Most hrHPV infections clear spontaneously, but around ten percent persist, with a smaller percentage progressing to CIN[3]. Factors like smoking, hormonal contraception, and parity may promote cancerous changes. Ongoing research explores various host, viral, genetic, and epigenetic factors in disease progression[4–17]. Recent studies also suggest a role for the local cervicovaginal microbiota (CVM) in influencing hrHPV infection and CIN risk[18–21].

The CMV plays an important role in reproductive health[18]. A healthy CVM exhibits low microbiota diversity, dominated by lactic acid-producing *Lactobacillus* species, which create an acidic environment that helps protect against infections and the overgrowth of pathobionts. The CVM can be classified into five community state types (CSTs) based on the abundance of specific Lactobacillus species or the presence of dysbiosis[22,23].

In CST I, II, III and V, dominant species are *Lactobacillus crispatus*, *Lactobacillus gasseri*, *Lactobacillus iners*, and *Lactobacillus jensenii*, respectively. In pre-menopausal women, oestrogen promotes the dominance of lactic acid-producing Lactobacillus species by upregulating glycogen production in the vaginal epithelium, which serves as a substrate for beneficial bacteria[24]. This supports immune homoeostasis by promoting anti-inflammatory responses and enhancing mucosal barrier integrity, reducing the likelihood of disease development[25].

Conversely, more diversity including anaerobic bacteria, such as *Gardnerella*, *Megasphaera*, *Atopobium*, *Sneathia*, or *Prevotella* associated with CST IV produce metabolites like putrescine and cadaverine, which neutralise lactic acid and weaken the protective acidic environment,

[1]Wolfson Institute of Population Health, Queen Mary University of London, London, UK. [2]The Institute of Cancer Research (ICR), Genomics Facility, Sutton, UK. [3]Wolfson Institute for Biomedical Research, University College London, London, UK. [4]Guy's and St. Thomas NHS Foundation Trust, London, UK. [5]These authors contributed equally: Cristiana Banila, Efthymios Ladoukakis. ✉e-mail: b.nedjai@qmul.ac.uk

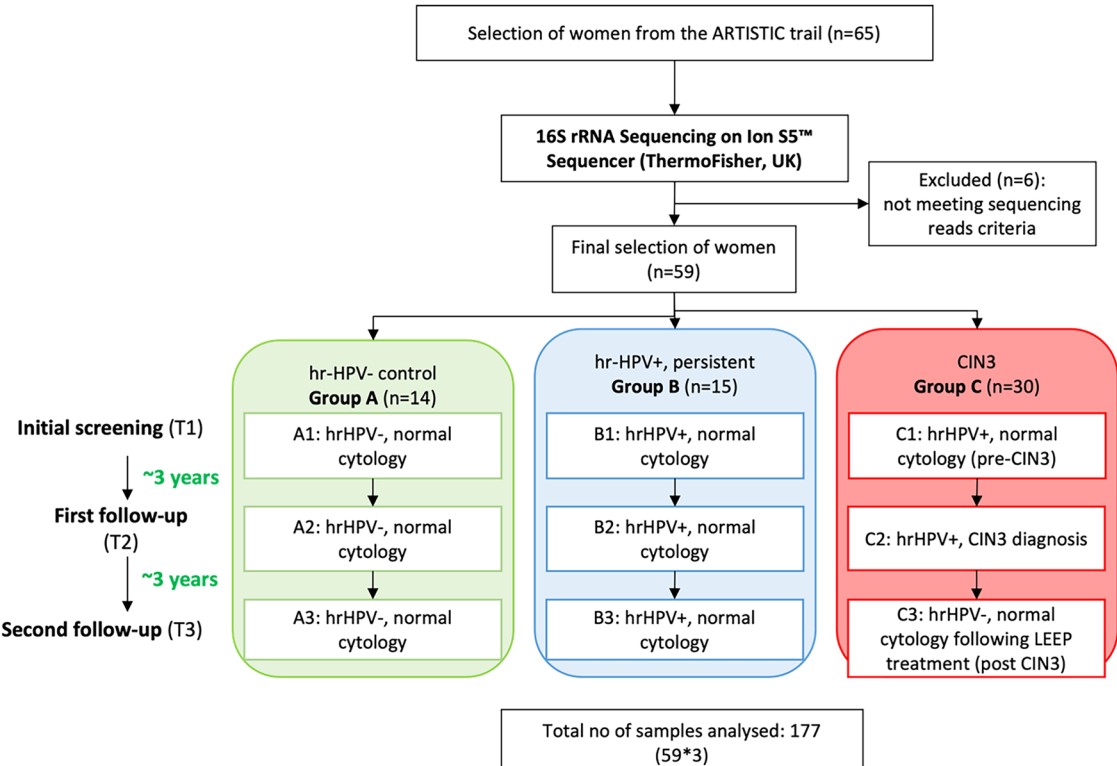

**Fig. 1 | Longitudinal microbiome study design flowchart.** A total of 65 women have been selected from the ARTISTIC trial. All women showed cytology negative results at enrolment and were followed up two times over the course of 6 years. In total, three cervical swabs were collected from each woman. All samples were prepared for 16S rRNA sequencing. Samples from six women were excluded due to not meeting sequencing reads criteria. Based on results after screening follow-up, the remaining 177 samples from 59 women were split into three analysis groups: Group A (hr-HPV- control); Group B (hr-HPV+ persistent); Group C (CIN3). Abbreviations: *hr-HPV-* high-risk human papillomavirus negative, *hr-HPV+* high-risk human papillomavirus positive, *CIN3* cervical intraepithelial neoplasia grade 3, *LEEP* loop electrosurgical excision procedure.

fostering conditions for HPV persistence and CIN3 progression[22,23,26]. Next generation sequencing studies of the CMV indicate that bacterial community structures are dynamic, being influenced by hormones and have a predisposition to become less stable during menstruation and more diverse during pregnancy[26]. The composition and stability of the vaginal microbiome are crucial factors influencing the host's innate immune response and vulnerability to infections and subsequently, disease. A recent longitudinal study revealed that a decrease in *Lactobacillus spp* is linked to persistent and progressive CIN2 infections at 12- and 24-month follow-ups, while *Lactobacillus* dominance increases the chances of disease regression[18]. Prior cross-sectional data also connect reduced *Lactobacillus spp* with increased anaerobic bacteria like *Sneathia, Gardnerella, Megasphaera, Atopobium,* or *Prevotella* to HPV and CIN persistence[19,27,28]. *Sneathia* species has been consistently associated with high-grade cervical lesions (e.g., CIN2/3) and hrHPV infection, suggesting its potential role as a microbial marker for progression[29]. Conversely, *Lactobacillus gasseri*-dominated microbiota is associated with faster hrHPV clearance, while lower *Lactobacillus* levels alongside high *Atopobium, Gardnerella,* and *Prevotella* abundance slow down HPV clearance[30,31]. CST IV is associated with bacterial vaginosis and has been linked to HPV, CIN, and cancer; however, it is unclear whether specific microbial signatures directly promote disease development or if the disease environment affects the microbial community.

Despite growing evidence linking cervical neoplasia driven by hrHPV infection to increased vaginal microbiota diversity, the long-term impact on vaginal microbiota after LEEP remains poorly understood. Limited studies have examined short-term changes post-excision, revealing a shift towards less diverse microbiota in Chinese and Norwegian studies at post-CIN excision[31,32]. A UK-led study observed that surgical excision of CIN3 led to a shift in CVM towards Lactobacillus-dominant states six months post-CIN3 excision[33]. Failure to re-establish a *Lactobacillus*-enriched CST may contribute to women remaining at high risk of pre-invasive and invasive disease recurrence. Therefore, we hypothesise that dysbiosis-associated anaerobes are promoting disease progression.

To test this, we designed a pilot longitudinal study with the primary objective to assess the long-term effects of CVM changes in pre-menopausal women three years after surgical excision. The second objective of the study is to identify microbial markers specific to CIN3 development compared to post-CIN3 excision, persistent hrHPV positive, and control specimens.

## Results

### Study design and patients' selection

Sixty-five women were included in the study, with cervicovaginal swabs taken at three time points, three years apart (Fig. 1). Samples from six women (*n* = 18) were excluded due to inadequate read coverage (<10,000 reads per sample). The samples from 59 women (samples, *n* = 177) were categorised into groups A, B, and C based on HPV type and cytology. Each group was subdivided into three subgroups according to initial screening (T1), first follow-up (T2) and second follow-up (T3). Group A (hrHPV-, control) included 14 women with negative HPV results and normal cytology (subgroups A1, A2 and A3). Group B (hrHPV+, persistent) consisted of 15 women consistently hrHPV+ with normal cytology, i.e., no CIN development (subgroups B1, B2 and B3). Group C (CIN3, progressive disease) comprised 30 women initially hrHPV+ with normal cytology i.e., no CIN development (subgroup C1/pre-CIN3), diagnosed with CIN3 at the first follow-up (subgroup C2/CIN3), treated surgically, and became hrHPV- with normal cytology results at second follow-up (subgroup C3/post-CIN3).

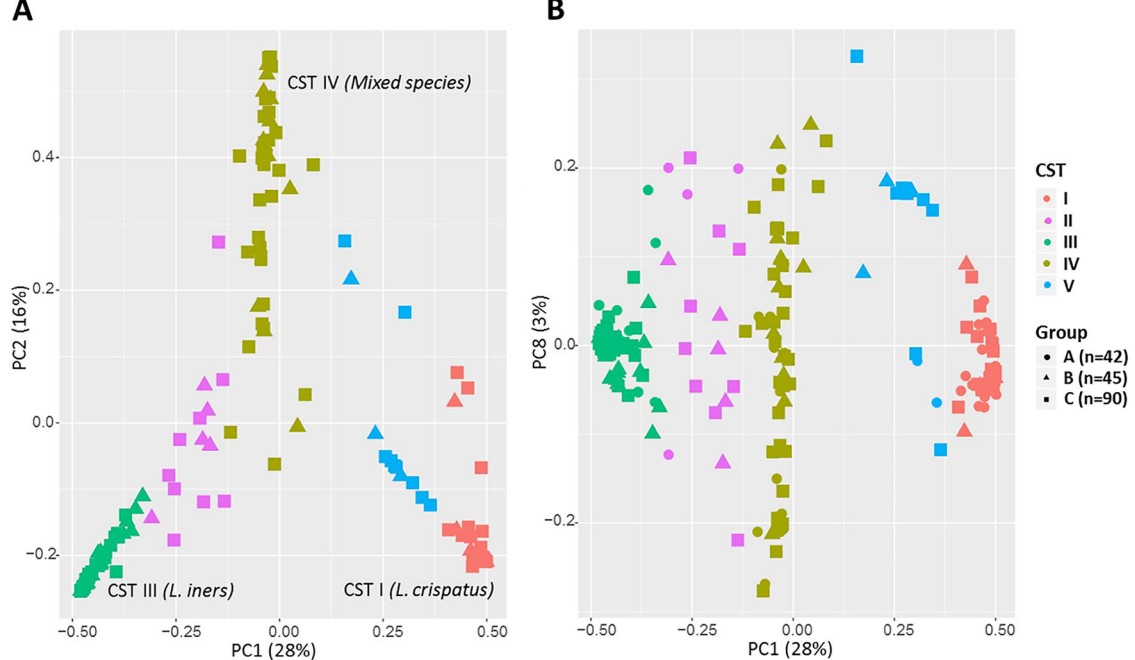

**Fig. 2 | Bacterial species beta diversity in the study.** Principal component analysis (PCA) based on Bray Curtis dissimilarities identified five major clusters matching to samples dominated by five previously described CSTs*: L. crispatus* (CST I), *L. gasseri* (CST II), *L. iners* (CST III), *L. jensenii* (CST V) and mixed species (CST IV) Samples belonging to different CSTs are indicated with different colours (**A**) PCA comparing

PC 1 to PC2 (**B**) PCA comparing PC1 to PC8 for more comprehensive view of the sample distribution. Abbreviations: *CST* community state type, *PC1* the first principal component, *PC2* the second principal component, *PC8* the eighth principal component.

## Cervico-vaginal microbiome composition by hrHPV positivity and disease status

A total of 15,846,102 reads were generated, averaging 89,529 reads per sample, with mean and median lengths of 232 and 259 bp, respectively, after barcode removal. Operational taxonomic units (OTUs) were assigned to each sample to mitigate sequencing bias, resulting in 236 OTUs, with a mean of 40.4 per sample. After removing singletons and rare OTUs, a total of 156 taxa remained. The top 37, identified as the most abundant taxa, accounted for 97.4% of the total reads. Subsequent analysis was limited to these top 37 taxa, with the remaining 119 taxa designated as "other".

CVM composition was analysed via principal component analysis (PCA), considering HPV status and cervical disease grade. Samples were grouped based on *Lactobacillus* abundance into *Lactobacillus*-dominant and *Lactobacillus*-depleted categories (<60% *Lactobacillus* abundance with higher anaerobic species diversity). Species-level classification revealed five clusters matching known CSTs: *L. crispatus*-dominant CSTI, *L. gasseri*-dominant CSTII, *L. iners*-dominant CSTIII, *Lactobacillus*-depleted with greater mixed species diversity CSTIV, and *L. jensenii*-dominant CSTV (Fig. 2). Hierarchical clustering analysis (HCA) confirmed the distinct clustering patterns observed in the groups (Fig. 3).

Table 1 summarises the distribution of CSTs among the analysed samples. In group A, representing hrHPV- women across all time points, CSTIII was the most frequent microbial state (50%). CSTI predominated in the group persistently hrHPV+ (group B) (42.2%). A larger number of samples with CSTI and CSTIV, and a significantly smaller number with CSTIII, was observed in group B compared to group A across all time points (*p* < 0.05).

Group C overall showed CSTIII as the most common (28.9%), with subgroup C1 (pre-CIN3) dominated by CSTI (33.3%) and resembling group B (hrHPV+ persistent) (42.2%). In subgroup C2 (CIN3), CSTIV most common (33.3%). Subgroup C3 (post-CIN3) showed CSTIII predominance (33.3%) and resembled group A (hrHPV-) in microbial state frequency.

CSTIII was more frequent in hrHPV- samples (43%) compared to hrHPV+ samples (21.9%). Conversely, CST I and IV were most common in hrHPV+ samples. CSTIV dominated in HPV16 positive samples (37.9%) and was most prevalent when compared to samples positive for HPV18 (12.5%) and other hrHPV types (25.8%).

## CVM diversity by hrHPV status and disease progression

Species richness analysis showed significantly higher number of species in samples classified as CSTIV compared to all other CSTs (Fig. 4A). Diversity was also significantly higher in CSTIV classified samples (*p* < 0.001) compared to CSTI, CSTII and CSTIII (Fig. 4B, C).

CVM diversity was higher in women with persistent hrHPV infection and during CIN3 progression (C1–C2). The highest richness and diversity was observed before and at CIN3 diagnosis, attributed to rising CSTIV rates in hrHPV+ samples (group B/hrHPV+ persistent—28.9% and subgroup C1/pre-CIN3—26.6%) and in CIN3 samples (subgroup C2—33.4%) (Table 1). The lowest CSTIV rate, and thus richness and diversity, was found in hrHPV- women, both in healthy (group A—19%) and post-CIN3 (subgroup C3—10%) groups. CIN3 surgical treatment impacted CVM composition, as seen in paired samples from women diagnosed with CIN3. The rate of women with CSTIV decreased from 33.4% to 10% (*p* < 0.05), while other *Lactobacillus*-dominated CST rates increased moderately by 5–10% (Table 1).

The distribution dynamics of CSTs pre- and post-CIN3 are illustrated in a Sankey plot (Fig. 5). Transitions between pre-CIN3 and CIN3 were seen in CSTs I–IV, but not CSTV. CSTIV correlates with higher disease severity, being most prevalent in CIN3. CSTs I,II and III were common after LEEP and hrHPV clearance. The main transition flux between CIN3 and post-CIN3 was observed among CST I–V. CST IV showed a trend of transitioning towards other CSTs.

## Identification of CVM markers for CIN3 development

LEfSe analysis highlighted significant overrepresentation of several microbial species in CIN3 diagnosed women (C2) compared to samples taken

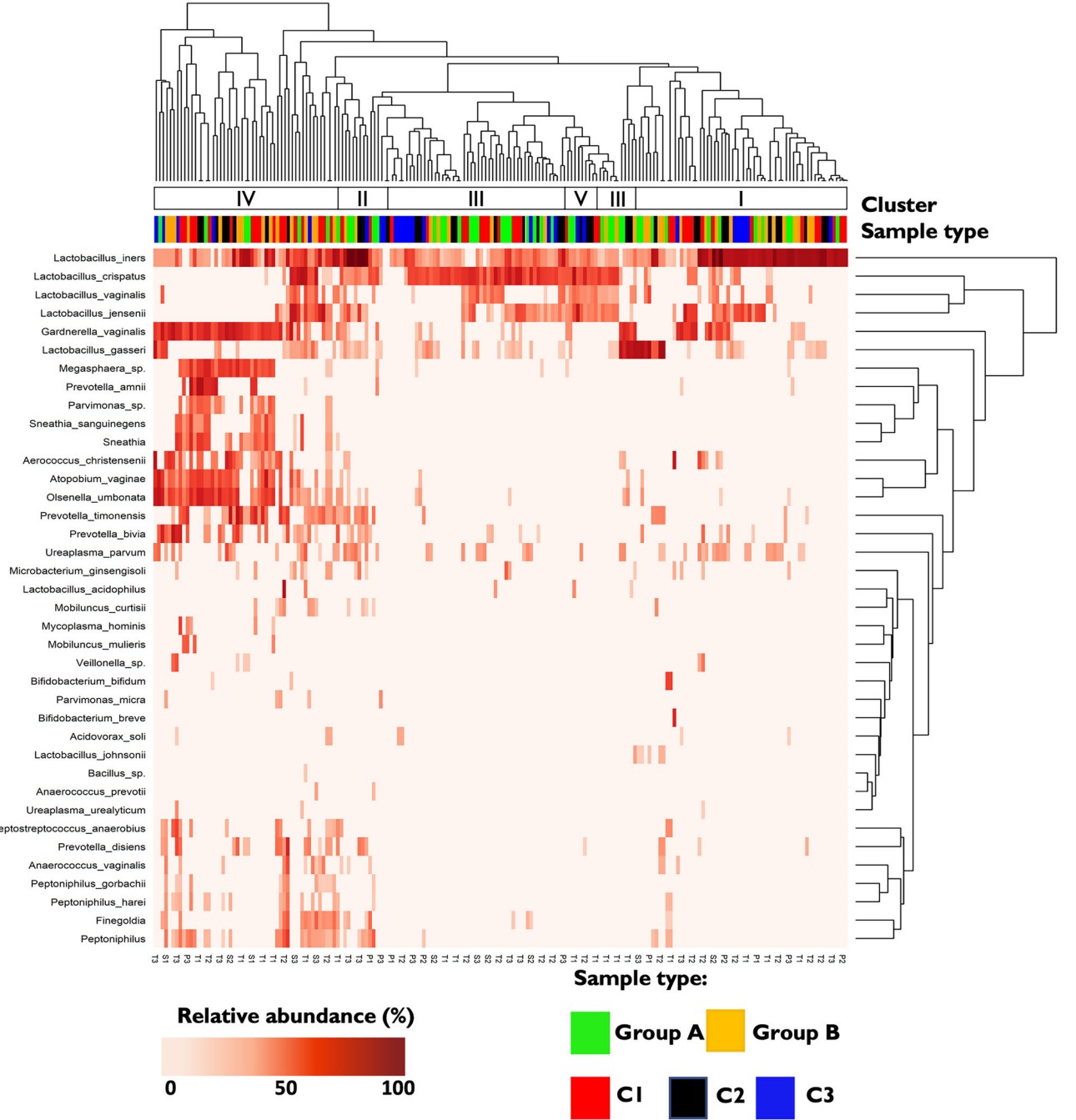

**Fig. 3 | Hierarchical clustering of 177 samples from 59 studied women.** The dendrogram was created using Euclidian centroid clustering of the 37 most abundant species determined in the samples. Each row represents a specific bacterial species, and each column represents a sample. Relative abundance of each species per sample is illustrated by the colour key. Sidebars above the heatmap represent: genus: clusters on genus level; community state type (CST): clusters on species level in compliance with previously described CSTs I-V; sample type based on women

HPV and disease status coded by colour: green - group A(hr-HPV negative with normal cytology, orange—group B (persistently hr-HPV+ and negative cytology), red—subgroup C (hr-HPV+ with negative cytology (progressing in the future to CIN3, (black – subgroup C2 (CIN3 diagnosed), blue – subgroup C3 (post- LEEP treatment, hr-HPV negative with normal cytology); hr-HPV status; hr-HPV genotype.

about 3 years before and after diagnosis (C1 and C3, respectively). Specifically, *Sneathia amnii* ($p < 0.01$), *Megasphaera genomosp.* ($p < 0.01$), *Peptosterptoccocus anaerobius* ($p < 0.05$), and *Achromobacter spanius* ($p < 0.05$) were identified. In contrast, post-CIN3 (C3) showed significant over-representation of *Lactobacillus* species, particularly *Lactobacillus gasseri* ($p < 0.01$), linked with hrHPV clearance (Fig. 6). Comparing persistently hrHPV+ samples (B3) with subgroup C2 identified *Sneathia amnii* as a significant differentiator, marking it as a microbial biomarker for CIN3 development ($p < 0.01$). Conversely, *Lactobacillus spp.*, including *L.*

*helveticus* ($p < 0.01$), *L. suntoryeus* ($p < 0.01$), and *L. vaginalis* ($p < 0.05$), were notably overrepresented in persistently hrHPV+ women (B3) compared to the CIN3 group (C2) (Supplementary Fig. 1).

## Discussion

This is a longitudinal cohort study to investigate differences in cervicovaginal bacterial composition between CIN3 progression, persistent hrHPV infection, and healthy controls, as well as the signature strains associated with CIN3 development and resolution. Specifically, we highlight

**Table 1 | Community states type's (CST's) distribution in 177 studied samples by sample type (group), hr-HPV status and HPV genotype**

| | CST I<br>*L. crispatus* n/N (%) | CST II<br>*L. gasseri* n/N (%) | CST III<br>*L. iners* n/N (%) | CST IV<br>*Mixed* n/N (%) | CST V<br>*L. jensenii* n/N (%) | Total<br>n/N |
|---|---|---|---|---|---|---|
| **Group A (hr-HPV-, normal cytology)** | | | | | | |
| A1 | 3/14 (21.4) | 2/14 (14.2) | 7/14 (50.0) | 2/14 (14.2) | 0/14 (0.0) | 14/14 (100) |
| A2 | 3/14 (21.4) | 0/14 (0.0) | 7/14 (50.0) | 3/14 (21.4) | 1/ 14 (7.1) | 14/14 (100) |
| A3 | 4/14 (28.5) | 0/14 (0.0) | 7/14 (50.0) | 3/14 (21.4) | 0/14 (0.0) | 14/14 (100) |
| Total group A | 10/42(23.8) | 2/42 (4.8) | 21/42(50.0) | 8/42 (19.0) | 1/42 (2.4) | 42/42 (100) |
| *P* value | 0.850 | | | | | |
| *Q* value | 0.850 | | | | | |
| **Group B (hr-HPV+, normal cytology)** | | | | | | |
| B1 | 6/15 (40.0) | 0/15 (0.0) | 2/15 (13.3) | 4/15 (26.6) | 3/15 (20.0) | 15/15 (100) |
| B2 | 6/15 (40.0) | 0/15 (0.0) | 4/15 (26.6) | 5/15 (33.3) | 0/15 (0.0) | 15/15 (100) |
| B3 | 7/15 (46.6) | 2/15(13.3) | 1/15 (6.6) | 4/15 (26.6) | 1/15 (6.6) | 15/15 (100) |
| Total group B | 19/45(42.2) | 2/45 (4.5) | 7/45 (15.5) | 13/45 (28.9) | 4/45 (8.9) | 45/45 (100) |
| *P* value | 0.449 | | | | | |
| *Q* value | 0.561 | | | | | |
| **Group C (progressive cervical disease)** | | | | | | |
| C1 (pre-CIN3) | 10/30 (33.4) | 3/30 (10.0) | 8/30 (26.6) | 8/30 (26.6) | 1/30 (3.4) | 30/30 (100) |
| C2 (CIN3) | 6/30 (20.0) | 6/30 (20.0) | 8/30 (26.6) | 10/30 (33.4) | 0/30 (0.0) | 30/30 (100) |
| C3 (post-CIN3) | 8/30 (26.6) | 7/30 (23.3) | 10/30 (33.4) | 3/30 (10.0) | 2/30 (6.7) | 30/30 (100) |
| *P* value | 0.338 | | | | | |
| *Q* value | 0.561 | | | | | |
| **hr-HPV status** | | | | | | |
| Negative (hr-HPV-) | 18/72 (25.0) | 10/72 (13.8) | 31/72 (43.0) | 11/72 (15.3) | 2/72 (2.7) | 72/72 (100) |
| Positive (hr-HPV+) | 35/105 (33.3) | 11/105 (10.5) | 23/105 (21.9) | 32/105(30.4) | 4/105 (3.8) | 105/105 (100) |
| Total group | 53/177 (29.9) | 21/177 (11.9) | 54/177(30.5) | 43/177(24.3) | 6/177 (3.4) | 177/177 (100) |
| *P* value | 0.016 | | | | | |
| *Q* value | 0.041 | | | | | |
| **hr-HPV genotype** | | | | | | |
| HPV16 | 19/58 (32.7) | 4/58 (6.9) | 13/58 (22.4) | 22/58 (37.9) | 0/58 (0.0) | 58/58 (100) |
| HPV18 | 10/16 (62.5) | 2/16 (12.5) | 2/16 (12.5) | 2/16 (12.5) | 0/16 (0.0) | 16/16 (100) |
| Other hr-HPV | 6/31 (19.4) | 5/31 (16.1) | 8/31 (25.8) | 8/31 (25.8) | 4/31 (12.9) | 31/31 (100) |
| Total group | 35/105(33.3) | 11/105 (10.5) | 23/105(21.9) | 32/105(30.4) | 4/105 (3.8) | 105/105 (100) |
| *P* value | 0.011 | | | | | |
| *Q* value | 0.041 | | | | | |

*CST* community state type, *CST I* = *Lactobacillus* crispatus-dominant, *CST II* = *Lactobacillus gasseri*-dominant, *CST III* = *Lactobacillus iners*-dominant, *CST IV* = high-diversity and *Lactobacillus spp*-depleted, *CST V* = *Lactobacillus jensenii*-dominant, *hr-HPV* high-risk human papilloma virus, *n* number of samples in a specified category, *N* total number of samples in a specified group. *P* values were calculated to assess subgroup similarities using Fisher's exact test, *Q* values were calculated using Benjamini-Hochberg false discovery rate (FDR) method.

Sneathia species as a potential biomarker for disease progression and Lactobacillus gasseri as an indicator of recovery and HPV clearance. To the best of our knowledge, this is the longest longitudinal study to date, spanning over six years and involving 59 reproductive-age women.

Our findings revealed a change in specific bacterial species when comparing CIN3, pre- and post-treatment, hrHPV persistent and healthy control samples. *Sneathia amnii* was strongly correlated to CIN3 development ($p \leq 0.05$), indicating its potential role in disease progression independent of hrHPV status, though further functional studies are needed to confirm causality. These findings reinforce the idea that cervical microbiota composition influences HPV persistence and CIN progression. If Sneathia amnii is consistently linked to CIN3 and persistent high-risk HPV (hrHPV) infections, it might serve as a microbial indicator of poor prognosis. Consistent with our findings, Zhang et al. noted an enrichment of *Sneathia amnii* pre-intervention compared to three months post-intervention in Chinese women with CIN2/3. Similarly, Mitra et al. found *Sneathia amnii*

and *Prevotella bivia* significantly higher before treatment compared to untreated controls. While *Sneathia amnii* decreased after surgical treatment, *Prevotella bivia* remained elevated, and *Lactobacillus crispatus* was less abundant compared to untreated controls[33]. These observations support the hypothesis that *Sneathia amnii* may indicate a cervical disease state.

Species from the genus *Sneathia* have emerged as pathogens in female reproductive diseases. While primarily associated with bacterial vaginosis, increasing evidence links *Sneathia* to cervical cancer progression. A study on the CVM of reproductive-age females found abundant *Sneathia species* (*S. amnii* and *S. sanguinegens*) in HSIL[34]. *Sneathia* was also correlated with HPV infection in a Korean twin cohort[35]. Łaniewski et al. demonstrated an increasing association between *Sneathia* abundance in HPV-positive specimens from low-grade and high-grade cervical lesions, compared to HPV-negative controls[19]. However, more longitudinal studies are needed to explore the associations between *Sneathia*, HPV and cervical disease. In vitro studies have shown that

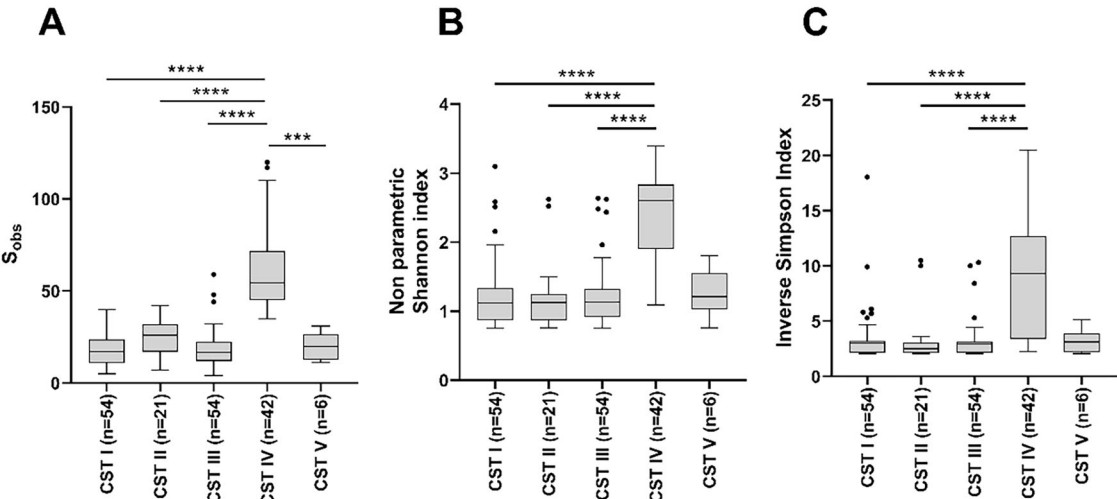

**Fig. 4 | Box-plot comparisons of bacterial species richness (A) and diversity (B, C) by CSTs identified in 177 studied samples.** A significantly higher species number was detected in CST IV samples compared to those classified as CST I ($P < 0.0001$), CST II ($P < 0.0001$), CST III ($P < 0.0001$) and CST V ($P < 0.001$) (**A**). Bacterial diversity using non-parametric Shannon alpha index (**B**) and Inverse Simpson index (**C**) was also significantly higher in CST IV samples comparing to CST I ($P < 0.0001$), CST II ($P < 0.0001$), CST III ($P < 0.0001$). Kruskall-Wallis and Dun's multiple comparisons tests were used to analyse data. Only significant differences were shown: ***=$P < 0.001$, ****=$P < 0.0001$. Abbreviations: *CST* community state type, $S_{obs}$ species observed.

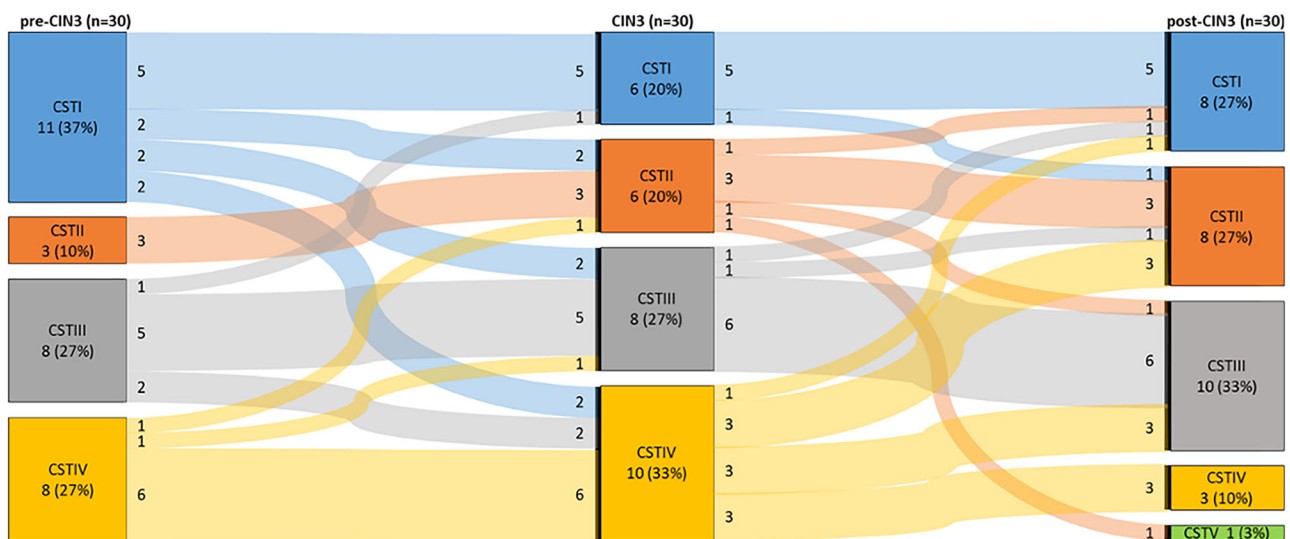

**Fig. 5 | Transition probabilities and stationary distribution of bacterial community state types (CST).** The transition probabilities were calculated using Sankey plot analysis for women in group C. The diagram describes a sequence of possible events in which the probability of each event depends only on the state attained in the previous event. Depending on the bacterial composition, four and five CSTs, respectively, have been identified to transition from one to another. The transitions between states are represented by arrows. The probability of one state transitioning into another is colour-coded. In the long run, the stationary distribution of CSTs showed CST IV as dominant in CIN3. CST I was found dominant in post-CIN3, followed by CST II and CST III. Analysis was performed in MATLAB Online R2019b. Abbreviations: *pre-CIN3* subgroup of C, including hrHPV women, *CIN3* subgroup of C, including CIN3 diagnosed women, *post-CIN3* subgroup of C, including women recovering post-LEEP treatment, *CST* Community state type.

*Sneathia amnii* significantly upregulates IL-1α, IL-β, TNF-α, and IL-8 in human vaginal epithelial cells. Specifically, TNF-α upregulation may damage the endocervical columnar epithelial barrier by disrupting tight junctions, facilitating bacterial and viral translocation and increasing susceptibility to viral acquisition[36].

Along with the association of cervicovaginal microbial imbalance and CIN3 progression, we observed a change in CVM state structure after LEEP treatment. Women diagnosed with CIN3 showed significantly higher levels of *Lactobacillus spp* depleted CVMs compared to post-CIN3 excision. This supports previous findings linking low *Lactobacillus spp* and increased anaerobic bacteria like *Sneathia, Gardnerella, Megasphera, Atopobium* or *Prevotella* with high-grade cervical disease[18,19,27,28]. *Sneathia* expression

remained elevated in women diagnosed with CIN3 (C2) compared to post-CIN3, persistent hrHPV+, and hrHPV- controls.

Our study demonstrated that CIN3 excision influences CVM composition, promoting transitions towards *Lactobacillus*-dominated states (CSTI, CSTII and CSTIII) three years post LEEP, when all women were hrHPV-. This aligns with previously reported data. Zhang et al. reported increased *Lactobacillus spp.* abundance three months post-treatment in 26 Chinese women with CIN2/3, including six post-menopausal individuals[37]. Similarly, Wiik et al. described a less diverse CVM in 89 Norwegian pre-menopausal women six and twelve months after CIN3 treatment[32]. In contrast, Mitra et al. did not observe significant *Lactobacillus spp.* enrichment six months post excision in 103 pre-menopausal women with high-

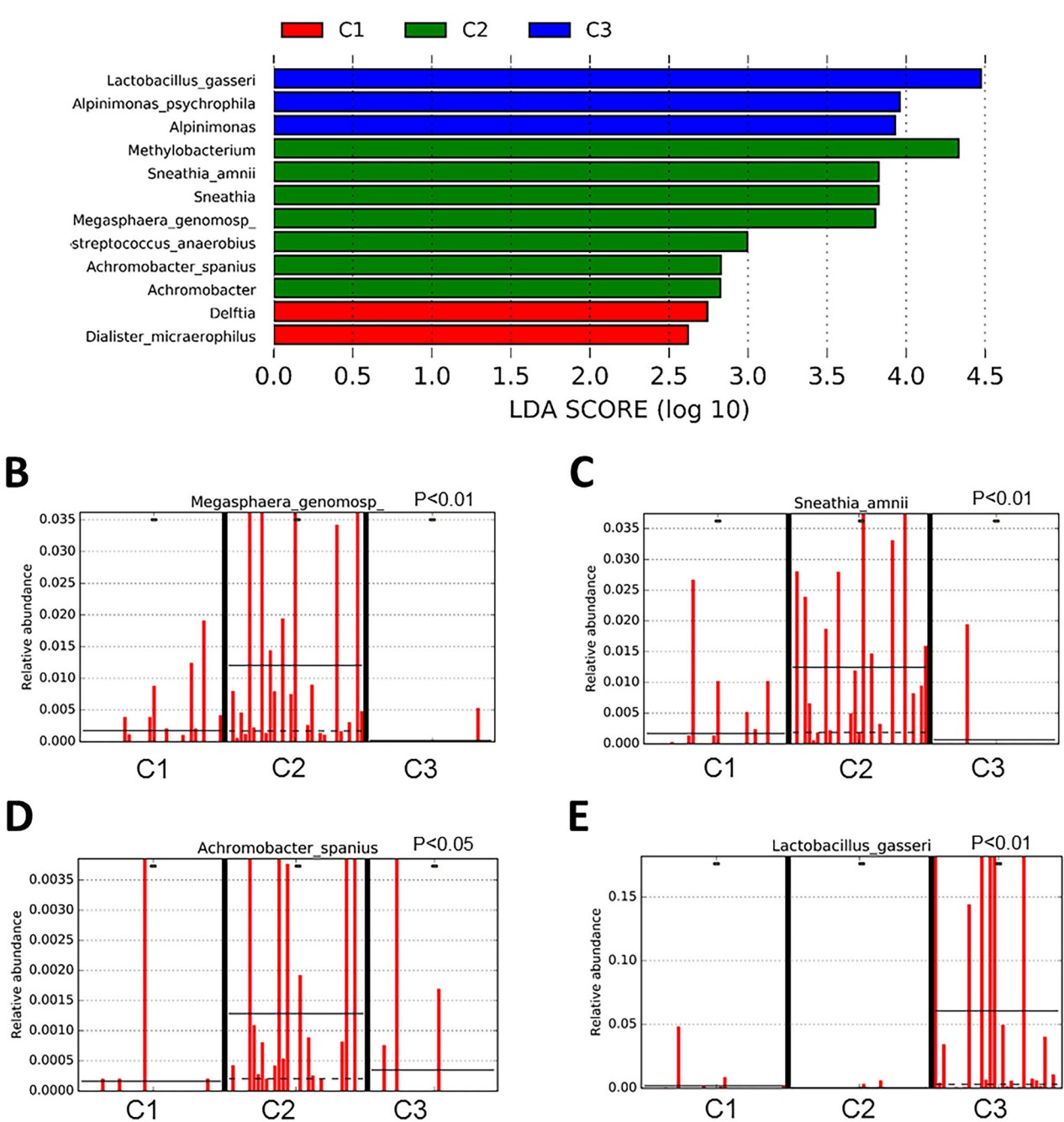

**Fig. 6 | Microbial biomarkers discovery in group C by LEfSe analysis.**
**A** Histogram of the LDA scores computed for differentially abundant taxa between sample subgroups of group C: C1 (pre-CIN3), C2 (CIN3) and C3 (post-CIN3). The length of the bar represents the log10 transformed LDA score, indicated by vertical dotted lines. The threshold on the logarithmic LDA score for discriminative features was 2.0. Relative abundance counts of *Megasphaera genomosp.* (**B**) *Sneathia amnii* (**C**) and *Achromobacter spanius* (**D**) significantly overrepresented in C2 samples,

while *Lactobacillus gasseri* (**E**) was significantly higher expressed in C3 samples. Differentially abundant sequences were determined using Welch's t-test. Abbreviations: C1, C2, C3 subgroups of group C (women with progressive CIN3 disease) who had samples taken 3 years before being diagnosed with CIN3 (C1/pre-CIN3), at the moment of diagnosis (C2/CIN3) and 3 years after LEEP-treatment (C3/post-CIN3). *LDA score* linear discriminant analysis score, *LEfSe* linear discriminant analysis effect size.

grade CIN[33]. They hypothesised that failure to re-establish a *Lactobacillus*-enriched CST may explain the continued high risk of disease recurrence. In our analysis, when we focused on the species level, we detected a higher abundance of *Lactobacillus gasseri* in post-CIN3 specimens compared to pre-CIN3 and control samples.

Beyond *Lactobacillus* gasseri, specific *Lactobacillus* species, including *Lactobacillus helveticus*, *Lactobacillus suntoryeus* and *Lactobacillus vaginalis*

(all $p \le 0.05$), were more abundant in women with persistent hrHPV positivity compared to those who developed CIN3. These findings suggest their protective role against CIN3 development, which can be attributed to their metabolomic profile. For instance, hydrogen peroxide-producing *Lactobacillus* strains possess bactericidal activity, inhibiting HPV invasion into cervical epithelial cells and preventing cervical lesions[21]. *Lactobacillus helveticus KS300* inhibits dysbiosis-associated anaerobic bacteria growth

and reduces their viability by impeding adhesion to cervical epithelial cells[38–40]. Moreover, *Lactobacillus helveticus* produces bioactive peptides with health benefits[41]. Consistent with our observation, *Lactobacillus helveticus* exhibits antibacterial properties in-vitro, but its full beneficial effects on the host require further investigation.

The relationship between cervicovaginal microbiome, inflammation and host immunity plays a crucial role in HPV clearance and CIN3 progression. Our study has demonstrated examples of such a role by reporting associations of *Sneathia* and *Lactobacillus gasseri* to diseased and recovered states, respectively. This is consistent with Brotman et al.'s findings, where *Lactobacillus gasseri* was linked to rapid HPV clearance in a cohort of 32 reproductive-age women[30]. The immune microenvironment is crucial in HPV clearance, with checkpoint proteins PD-L1 and LAG-3 positively correlating with dysbiosis-associated bacteria like *Sneathia* and *Prevotella*, and LAG-3 negatively correlating with *Lactobacillus gasseri*. This suggests that these bacteria can modulate cervicovaginal immune responses and host susceptibility to infection[36]. These species could indicate recovery and HPV clearance, but it remains unclear if distinct microbial signatures promote disease development or regression.

The current study's limitations include a lack of ethnic diversity and individual epidemiological data for each woman. The lack of ethnic diversity in the study cohort could mask or overestimate the impact of specific microbial signatures on CIN3 progression or regression. Factors like recent sexual activity, contraceptive use or menstrual cycle information, which could influence results, were unknown due to the reliance on retrospective data collection in the ARTISTIC cohort, which did not record these[14,42]. The CVM typically exhibits the highest diversity and instability during menstruation, when oestrogen and progesterone levels are lowest. Although the exact timing of the last menstruation was unknown, none of the samples were collected during menstruation. Furthermore, despite being longitudinal in design, the study had a modest sample size for all analysed groups, emphasising the need for more extensive data collection to achieve statistically significant results.

In summary, this pilot longitudinal study revealed that women diagnosed with CIN3 had significantly higher levels of *Lactobacillus spp*-depleted CVMs compared to post-CIN3. The higher presence of *Lactobacillus gasseri* in post-CIN3 specimens aligns with previous findings associating this species with faster HPV clearance. This suggests a potential protective or recovery-promoting role of Lactobacillus gasseri in cervical health. Additionally, *Sneathia amnii* consistently appeared as a potential microbial biomarker for CIN3 development, exhibiting differential abundance in CIN3 compared to post-CIN3 specimens, persistently hrHPV+, and control groups. Further research is needed to elucidate the connection between the identified microbial signature and its impact on epigenetic mechanisms. Conversely, higher proportions of *Lactobacillus helveticus*, *Lactobacillus suntoryeus*, and *Lactobacillus vaginalis* demonstrated a potential protective role against CIN3 development in women with persistent hrHPV infections. Microbial biomarkers such as Sneathia amnii and Lactobacillus gasseri could complement HPV screening to refine risk stratification. Future studies should explore whether modulating the vaginal microbiota via probiotics or targeted antimicrobial therapy may enhance HPV clearance and prevent CIN3 recurrence.

## Methods
### Study population—inclusion and exclusion criteria
Ethics approval was obtained in the UK with the REC reference: 14/LO/0627 IRAS project ID: 153311 for the ARTISTIC trial. Sixty-five women were selected from the trial, which compared screening outcomes using cervical cytology alone versus cytology plus HPV testing over two rounds, three years apart, as described in detail previously[42]. All patients gave informed consent. All ethical regulations relevant to human research participants were followed. We included women pre-menopausal (20–53 years), non-pregnant, with hrHPV genotyping results available for each screening round. Women were included irrespective of their ethnicity, parity, smoking habits, phase in their cycle and use of contraception. Women who were HIV

or hepatitis B/C positive, with autoimmune disorders, who received antibiotics or pessaries within 14 days of sampling, or had a previous history of cervical treatment were excluded. Ethnicity was self-reported as Caucasian. Detailed medical history can be found in the ARTISTIC trial[42]. Samples from six women (samples, $n = 18$) were excluded due to inadequate read coverage (<10,000 reads per sample or ≥10x coverage in at less than 70% of reads). The final analysis looked at 59 women (samples, $n = 177$).

### Specimen collection and processing
Cervico-vaginal samples were processed for cervical cytology and HPV genotyping based on ARTISTIC trial's protocols[42]. Residual samples were centrifuged, supernatant removed, and pellets resuspended in 800 μl sterile PBS, transferred to 2 ml vials and stored at –80 °C. Bacterial DNA was extracted from 500 μl of PBS-suspended cervical samples using the DNeasy Blood&Tissue Kit (Qiagen, Hilden, Germany) per the manufacturer's instructions. DNA quantification was performed with Qubit, and samples were stored at −80 °C for further analysis.

### Metagenomic sequencing
Extracted DNA was processed using the Ion 16S™ Metagenomics Kit. Each sample underwent two PCR reactions targeting hypervariable regions of the 16S rRNA gene (V2-4-8 for PSI and V3-6, 7–9 for PSII) with specific primers. The resulting PSI and PSII PCR products were combined and purified using Agencourt® AMPure XP® Magnetic Beads. 50 ng of each purified PCR product was included in the final pool. Subsequent libraries were prepared using the Ion Plus Fragment Library Kit and unique barcoded primers. The Ion Universal Library Quantification Kit was used to quantify the purified libraries. For template preparation, an input of 2.5 pM per library was added on the Ion Chef™ Instrument using the Ion 510™/520™ 530™ Kits – Chef, according to the manufacturer's instructions. Subsequently, 16S rRNA metagenomic sequencing was performed on the Ion S5™ Sequencer following the manufacturer's protocol. All reagents were sourced from ThermoFisher, UK. Multiple sequencing runs were performed per sample to ensure sufficient sequencing depth, improve coverage uniformity, and mitigate technical variability. In some instances, initial runs did not meet the required quality thresholds or read depth (≥10,000 reads per sample and ≥10x coverage in at least 70% of reads), necessitating additional sequencing. This approach also provides redundancy, allowing for more robust downstream analysis after merging reads from multiple runs. Sample names in Supplementary Data 1 include information on the sequencing runs performed in the project. Suffixes such as '_merged' in the sample names indicate three sequencing runs have been merged together in a file, '_v1' indicate only one sequencing run has been used for analysis. No suffix on the sample indicates only two sequencing runs merged together for subsequent analysis.

### Metagenomic sequence processing and quality control
Raw FASTQ files underwent quality assessment using FastQC, followed by adaptor trimming and quality filtering with Trimmomatic v0.43[43], applying a threshold of ≥10x coverage in at least 70% of targeted sites. To ensure reliable taxonomic and diversity analysis, only samples with ≥10,000 high-quality reads were retained for downstream processing.

Samples passing this threshold were aligned against two reference databases for microbial taxonomic classification: the MicroSEQ® ID 16S rRNA database (ThermoFisher; containing >15,000 curated bacterial entries) and the GreenGenes 16S rRNA gene database (version 13_8; containing >400,000 sequences)[44]. Alignments were performed using Ion Reporter™ Software v4.2 (ThermoFisher, UK).

Reads were normalised to the lowest sample read count to allow fair comparisons across samples. The resulting alignment files (SAM format) were converted to sorted and indexed BAM files using Samtools v1.3, then uploaded to the Harvard's University Galaxy Bioinformatics platform[45] for downstream processing.

For taxonomic and diversity analyses, OTUs were defined using a 98% sequence similarity threshold in Ion Reporter™. Linear discriminant analysis

effect size (LefSe) was applied for supervised clustering and identification of taxa differentially abundant between groups. Microbial profiles were assessed from phylum to species level, and both Alpha (within-sample) and Beta (between-sample) diversity indices were computed using classifications from both reference databases (Supplementary Data 1).

## Statistics and reproducibility

Taxonomic $\alpha$-diversity was estimated by OTUs count clustering at 97% similarity thresholds, deriving non-parametric Shannon and Inverse Simpson indices. Differences between groups were tested using the Wilcoxon rank-sum test. This is a non-parametric test used to compare differences between two independent groups. It ranks data points rather than comparing means, allowing for robust comparisons of microbial diversity, species abundance, or other continuous variables between groups (i.e., CST composition in pre-treatment vs. post-treatment groups). Data underwent multivariate analysis, including HCA using Euclidean centroid clustering of the 37 most abundant species with a clustering density threshold of 0.75. $P$ values and $Q$ values were calculated using Fisher's exact test and Benjamini-Hochberg false discovery rate (FDR) method, respectively. Fisher's exact test is specifically designed for small sample sizes and is used to assess the association between two categorical variables, such as comparing the presence or absence of a specific microbial species between different groups (i.e., pre-treatment vs. post-treatment groups or hrHPV+ vs. hrHPV− samples). PCA based on Bray-Curtis indices was used to visualise differences in microbial community structure according to CSTs distribution. Bray-Curtis dissimilarity was presented as the distance measure, where 0 represents identical samples and 1 represents communities that do not share any species. All $p$ values were two-sided with $\alpha \leq 0.05$ considered significant and adjusted for FDR. Analysis was performed using the 'vegan' package in R. LefSe analysis was conducted to find significant differences in the relative abundance of taxa within and between the investigated groups, using default values of $\alpha = 0.05$ and LDA = 2. Analysis was performed using publicly available software on Harvard University's Galaxy instance[45].

## Ethics statement

The use of the ARTISTIC samples in this study have been approved by the NRES Committee Southeast Coast – Brighton and Sussex (Study title: Long-term follow-up of ARTISTIC cervical screening trial cohort) with the REC reference: 14/LO/0627 IRAS project ID: 153311.

## Reporting summary

Further information on research design is available in the Nature Portfolio Reporting Summary linked to this article.

## Data availability

The OTU table created and used for each analytical step is included as Supplementary Data 1. The 16 s rDNA raw sequencing data for this study have been deposited in the Genome Sequence Archive (https://ngdc.cncb.ac.cn/gsa/) under the BioProject PRJCA039846.

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

## Acknowledgements

We extend our gratitude to C.G. and J.P. from the Non-Communicable Disease Epidemiology Unit at the London School of Hygiene & Tropical Medicine, London, UK, for their invaluable assistance with sample selection and metadata delivery. This work was supported by CRUK project grant with reference: C27045/A27046. We gratefully acknowledge their financial support.

## Author contributions

C.B., C.R. and B.N. conceived and designed the study. C.B. and M.K. performed experiments. B.N., C.B., D.S-B., E.L. and L.R.S. analysed the data. C.B., E.L., D.S.-B. and B.N. wrote the manuscript.

## Competing interests

The authors declare no competing interests.
