## [Transparent Peer Review file · Communications Biology]

A longitudinal pilot study in pre-menopausal women links cervicovaginal microbiome to CIN3 progression and recovery

Corresponding Author: Dr Belinda Nedjai

Version 0:

Reviewer comments:

Reviewer #1

(Remarks to the Author)

This is a longitudinal cohort study to investigate differences in vaginal bacterial composition between negative controls, persistent hrHPV infection, and pathological CIN3, as well as the signature strains associated with CIN3 development and resolution. The longitudinal cohort observation study is of great value in the field of vaginal microbiome, which is rarely studied. However, the paper has the following problems need to be improved.

Major revision

1. The title could be more specific about the study's longitudinal nature and its focus on premenopausal women.
1. It is suggested that the abstract emphasize the clinical characteristics of the study object: non-menopausal period (childbearing age) and LEEP treatment for CIN3.
2. Highlight the hypotheses explicitly in the introduction for better context. Why are specific microbial markers expected to correlate with CIN3 progression or recovery?
3. Emphasize the importance of studying premenopausal women explicitly to address hormonal influences on microbiota in the introduction.
4. Provide more detail about inclusion and exclusion criteria for women. Were confounding factors like sexual activity or contraception use considered?
5. Clarify if all participants were in the same phase of their menstrual cycle during sample collection.
6. The group introduction is not detailed and accurate. What are the pathological results of Group B patients? Is there CIN1/2? Is the pathological result of Group C1 negative?
7. Ensure consistent use of terms such as "CIN3 recovery," "post-treatment," and "microbial biomarkers." For example, define whether "post-CIN3" consistently means post-treatment and hrHPV-negative.
8. Explain why certain statistical tests, such as Fisher's exact test and Wilcoxon rank-sum test, were chosen for specific comparisons.
9. For Group C (CIN3), clarify the discrepancies noted between the text and Figure 1 regarding the HPV-negative and HPV-positive post-treatment groups.
10. Figure 3 Illustration is not clear and difficult to read and understand.
11. Clearly articulate the implications of identifying *Sneathia amnii* and *Lactobacillus gasseri* as biomarkers for disease progression and recovery.
12. Compare the results with similar studies, emphasizing unique contributions (e.g., longest longitudinal design, premenopausal focus).
13. Limitations: Discuss the lack of ethnic diversity in more depth and how it may limit generalizability. Address why epidemiological data like sexual activity, contraceptive use, and detailed menstrual cycle information were unavailable.
14. Ensure references are up-to-date, particularly in rapidly evolving microbiome research fields.

Reviewer #2

(Remarks to the Author)

Comments:

This longitudinal pilot study investigates vaginal microbiome signatures associated with CIN3 development and recovery over a 6-year period. The study analyzed samples from 59 pre-menopausal women, examining microbiota composition

changes before CIN3 development, during disease, and post-treatment.

The study is well-structured with a clear methodology using 16S rRNA sequencing and comprehensive statistical analyses. As a proof-of-concept study, the results are thoroughly presented with appropriate statistical support in tables and figures. The discussion effectively contextualizes the findings within existing literature. Study limitations are appropriately acknowledged.

There are very few minor comments:

Abstract:

Consider adding p-values for key findings if possible

Methods:

- Line 104: Please add "samples": six women (samples, n=18)

Results:

- For lines 267-270: Please add p-values for the reported changes in CST IV rates (from 33.4% to 10%) and the increases in Lactobacillus-dominated CST rates

Version 1:

Reviewer comments:

Reviewer #1

(Remarks to the Author)

The authors have revised the review comments one by one, but there are still the following points that need to be modified:

1. It is suggested to change the title of the article from "vaginal microbiome" to "cervicovaginal microbiome".
2. The discussion section needs to be revised.

Line 355-424 contains a large amount of information and scattered contents. To enhance the logic and readability of the article, it is recommended to divide the discussion in the following order: vaginal microbial imbalance and CIN3 progression, characteristics of microbial recovery after LEEP (emphasis on *Lactobacillus gasseri*), protective effect of *Lactobacillus* strains, inflammation and microbial interaction mechanism (possible mechanism of *Sneathia amnii* mediating CIN3 progression).

"*Sneathia amnii* was strongly associated to CIN3 development.....hrHPV." (line 389-391) change to "*Sneathia amnii* was strongly correlated with CIN3 development, indicating its potential role in disease progression independent of hrHPV status, though further functional studies are needed to confirm causality."

"Microbial biomarkers like *Sneathia amnii* and *Lactobacillus gasseri* could refine risk stratification for HPV-positive patients." (line 448-450) change to "Microbial biomarkers such as *Sneathia amnii* and *Lactobacillus gasseri* could complement HPV screening to refine risk stratification. Future studies should explore whether modulating the vaginal microbiota via probiotics or targeted antimicrobial therapy may enhance HPV clearance and prevent CIN3 recurrence."

Reviewer #2

(Remarks to the Author)

I would like to thank the authors for addressing my comments thoroughly and appropriately. Their responses demonstrate a commitment to improving the manuscript's quality, and the changes they've made enhance the clarity of the work. I am satisfied with how they have addressed each point, and I have no further concerns regarding the manuscript. I support the publication of this paper in its current form.

Responses to Reviewer 1

Dear Reviewer,

Thank you very much for your time and effort in reviewing our manuscript “A longitudinal pilot study in pre-menopausal women links vaginal microbiome to CIN3 progression and recovery” (title changed from: “Vaginal Microbiome Signatures Linked to CIN3 Progression and Recovery: Insights from a Longitudinal Pilot Study”) that is currently under consideration for publication in Communications Biology. Please find below our responses to your comments:

1. The title could be more specific about the study's longitudinal nature and its focus on premenopausal women.

The title is now changed to “A longitudinal pilot study in pre-menopausal women links vaginal microbiome to CIN3 progression and recovery” addressing the inclusion of pre-menopausal women as well as the longitudinal nature of the study.

2. It is suggested that the abstract emphasizes the clinical characteristics of the study object: non-menopausal period (childbearing age) and LEEP treatment for CIN3.

The abstract has been amended to emphasise the clinical characteristic of the study object. The fact that the women included in the study are non-menopausal as well as the LEEP treatment used for CIN3 are now stated clearly.

3. Highlight the hypotheses explicitly in the introduction for better context. Why are specific microbial markers expected to correlate with CIN3 progression or recovery?

The introduction has been amended to highlight our hypothesis of specific microbial markers expected to correlate with CIN3 progression. More diversity including anaerobic bacteria, associated with CST IV, produce metabolites that foster conditions for HPV persistence and CIN3 progression. We describe how this hypothesis is backed up by the relevant literature references (lines 65-89).

4. Emphasize the importance of studying premenopausal women explicitly to address hormonal influences on microbiota in the introduction.

The importance of studying premenopausal women explicitly to address hormonal influences on microbiota is now emphasised in the introduction (lines 68-79). In pre-menopausal women, oestrogen promotes the dominance of lactic acid-producing *Lactobacillus* species which promote anti-inflammatory responses. Anaerobic

bacteria, associated with CST IV produce metabolites that neutralise lactic acid and weaken the protective acidic environment, fostering conditions for HPV persistence and CIN3 progression. NGS studies of the CMV indicate that bacterial community structures are dynamic, being influenced by hormones and have a predisposition to become less stable during menstruation and more diverse during pregnancy.

5. Provide more detail about inclusion and exclusion criteria for women. Were confounding factors like sexual activity or contraception use considered?

The Materials and Methods section has been amended to include a more detailed explanation about the inclusion and exclusion criteria (lines 111-123). The subsection "Study population" has been renamed to "Study population – Inclusion and Exclusion criteria".

6. Clarify if all participants were in the same phase of their menstrual cycle during sample collection.

Women were included irrespective of their ethnicity, parity, smoking habits, phase in their cycle and use of contraception. This is now mentioned in the Methods section (lines 117-118).

7. The group introduction is not detailed and accurate. What are the pathological results of Group B patients? Is there CIN1/2? Is the pathological result of Group C1 negative?

Additional details about the groups B and C as well as Group B's pathological results are now included in the Results section (lines 191-196)

8. Ensure consistent use of terms such as "CIN3 recovery," "post-treatment," and "microbial biomarkers." For example, define whether "post-CIN3" consistently means post-treatment and hrHPV-negative.

This has been addressed throughout the manuscript.

9. Explain why certain statistical tests, such as Fisher's exact test and Wilcoxon rank-sum test, were chosen for specific comparisons.

An explanation was added in the "Statistics and Reproducibility" section (renamed from Statistical analysis) in lines 163-167 and 170-174. Wilcoxon rank-sum test is a non-parametric test used to compare differences between two independent groups allowing for robust comparisons of microbial diversity, species abundance, or other continuous variables between groups (i.e. CST composition in pre-treatment vs. post-treatment groups). Fisher's exact test is specifically designed for small sample sizes and is used to assess the association between two categorical variables i.e. such as comparing the presence or absence of a specific microbial species between

different groups (i.e. pre-treatment vs. post-treatment groups or hrHPV+ vs. hrHPV– samples).

10. For Group C (CIN3), clarify the discrepancies noted between the text and Figure 1 regarding the HPV-negative and HPV-positive post-treatment groups.

Figure 1 was amended to better reflect the different subgroups in Group C (see below). The Results section was also amended to better explain the Group C subgroups (lines 193-196)

11. Figure 3 Illustration is not clear and difficult to read and understand.

Figure 3 was remade in better quality (see below). Extra details were added to the legend (lines 232-240).

12. Clearly articulate the implications of identifying *Sneathia amnii* and *Lactobacillus gasseri* as biomarkers for disease progression and recovery. ->

The implications of identifying *Sneathia amnii* and *Lactobacillus gasseri* as biomarkers for disease progression and recovery have now been included in the discussion (lines 391-394, 438-441 and 448-450). We describe that since *Sneathia amnii* is consistently linked to CIN3 and persistent high-risk HPV (hrHPV) infections, it might serve as a microbial indicator of poor prognosis. Additionally, we mention that the higher presence of *Lactobacillus gasseri* in post-CIN3 specimens aligns with previous findings and suggests a potential protective or recovery-promoting role of L.

gasseri in cervical health. Finally, we conclude that the integration of microbial biomarkers like *Sneathia amnii* and *Lactobacillus gasseri* into screening or treatment strategies could refine risk stratification for HPV-positive patients, leading to more personalized interventions.

13. Compare the results with similar studies, emphasizing unique contributions (e.g., longest longitudinal design, premenopausal focus).

The Discussion section now includes comparison with similar studies (lines 363-366, 370-381 and 394-400). We denote that to our knowledge this is the longest longitudinal study to date (lines 359-360).

14. Limitations: Discuss the lack of ethnic diversity in more depth and how it may limit generalizability. Address why epidemiological data like sexual activity, contraceptive use, and detailed menstrual cycle information were unavailable. ->

The potential limitations of the study are now referenced in the Discussion section (lines 425-436). We mention that factors like recent sexual activity, contraceptive use or menstrual cycle information were unknown due to the reliance on retrospective data collection in the ARTISTIC cohort, which did not record these.

15. Ensure references are up-to-date, particularly in rapidly evolving microbiome research fields.

References throughout the text were updated and more contemporary references were included when needed.

Dear Reviewer,

Thank you very much for your time and effort in reviewing our manuscript “A longitudinal pilot study in pre-menopausal women links vaginal microbiome to CIN3 progression and recovery” (title changed from: “Vaginal Microbiome Signatures Linked to CIN3 Progression and Recovery: Insights from a Longitudinal Pilot Study”) that is currently under consideration for publication in Communications Biology. Please find below our responses to your comments:

Abstract:

1. Consider adding p-values for key findings if possible

P-values are now added for key findings both in the abstract (line 36) and in the Results section (lines 298 and 319-328)

2. Methods: Line 104: Please add “ samples”: six women (samples, n=18)

This is now added (new line: 122)

3. Results: For lines 267-270: Please add p-values for the reported changes in CST IV rates (from 33.4% to 10%) and the increases in Lactobacillus-dominated CST rates

This is now added (new lines: 298-299)

Responses to Reviewer 1

Dear Reviewer,

Thank you very much for your time and effort in reviewing our manuscript “A longitudinal pilot study in pre-menopausal women links cervicovaginal microbiome to CIN3 progression and recovery” (title changed from: “A longitudinal pilot study in pre-menopausal women links vaginal microbiome to CIN3 progression and recovery”) that is currently under consideration for publication in Communications Biology. Please find below our responses to your comments:

1. It is suggested to change the title of the article from "vaginal microbiome" to "cervicovaginal microbiome".

The title is now changed to “A longitudinal pilot study in pre-menopausal women links cervicovaginal microbiome to CIN3 progression and recovery”.

2. The discussion section needs to be revised. Line 355-424 contains a large amount of information and scattered contents. To enhance the logic and readability of the article, it is recommended to divide the discussion in the following order: vaginal microbial imbalance and CIN3 progression, characteristics of microbial recovery after LEEP (emphasis on *Lactobacillus gasseri*), protective effect of *Lactobacillus* strains, inflammation and microbial interaction mechanism (possible mechanism of *Sneathia amnii* mediating CIN3 progression).

The Discussion section is now revised as per reviewer 1's suggestion. More specifically:

- vaginal microbial imbalance and CIN3 progression is described in lines 197-222
- characteristics of microbial recovery after LEEP (emphasis on *Lactobacillus gasseri*) are described in lines 223-241
- the protective effect of *Lactobacillus* strains is described in lines 242-253
- inflammation and microbial interaction mechanism (describing the possible mechanism of *Sneathia amnii* mediating CIN3 progression) is described in lines 254-264

3. "Sneathia amnii was strongly associated to CIN3 development.....hrHPV." (line 389-391) change to "Sneathia amnii was strongly correlated with CIN3 development, indicating its potential role in disease progression independent of hrHPV status, though further functional studies are needed to confirm causality."

The suggested change was made (new lines 198-201)

4. "Microbial biomarkers like Sneathia amnii and Lactobacillus gasseri could refine risk stratification for HPV-positive patients."(line 448-450) change to "Microbial biomarkers such as Sneathia amnii and Lactobacillus gasseri could complement HPV screening to refine risk stratification. Future studies should explore whether modulating the vaginal microbiota via probiotics or targeted antimicrobial therapy may enhance HPV clearance and prevent CIN3 recurrence."

The suggested change was made (new lines 288-291)

Responses to Reviewer 2

1. I would like to thank the authors for addressing my comments thoroughly and appropriately. Their responses demonstrate a commitment to improving the manuscript's quality, and the changes they've made enhance the clarity of the work. I am satisfied with how they have addressed each point, and I have no further concerns regarding the manuscript. I support the publication of this paper in its current form.

Dear Reviewer,

Thank you very much for your time and effort in reviewing our manuscript "A longitudinal pilot study in pre-menopausal women links cervicovaginal microbiome to CIN3 progression and recovery" (title changed from: "A longitudinal pilot study in pre-menopausal women links vaginal microbiome to CIN3 progression and recovery") that is currently under consideration for publication in Communications Biology. We appreciate your positive feedback and your support for the publication of our work.